# Probability-Based City-Scale Risk Assessment of Passengers Trapped in Elevators under Earthquakes

**Donglian Gu [1], Yixing Wang [2], Xinzheng Lu [3] and Zhen Xu [1,*]**

[1] Research Institute of Urbanization and Urban Safety, School of Civil and Resource Engineering, University of Science and Technology Beijing, Beijing 100083, China

[2] Institute of Engineering Mechanics, China Earthquake Administration, Harbin 150000, China

[3] Key Laboratory of Civil Engineering Safety and Durability of China Education Ministry, Department of Civil Engineering, Tsinghua University, Beijing 100084, China

[*] Correspondence: xuzhen@ustb.edu.cn

**Abstract:** An accurate prediction of the number of passengers trapped in elevators under earthquakes in urban areas is essential for promoting earthquake emergencies. A probability-based city-scale method for assessing the earthquake-induced risk of passenger entrapment in elevators was proposed, in which city-scale time history analysis was performed to simulate the seismic response of building clusters, and the Monte Carlo simulation was conducted to consider the uncertainty of multiple factors, including the mechanical properties of buildings and elevators, the elevator story position, and the spatiotemporal nature of elevator traffics. A case study of the Tsinghua University campus was performed to demonstrate the practicability of the method. The results show that the number of trapped passengers when an earthquake occurs during the off-peak hours of elevator traffic is approximately a quarter of that when the earthquake occurs at 8:00. The maximum number of trapped passengers under the maximum considered earthquake reaches 195, approximately five times that under the design basis earthquake. This study fills a gap in the research on city-scale earthquake-induced passenger entrapment risk. The proposed method can be used to perform both scenario- and intensity-based assessments, thereby having the potential to facilitate virtual rescuer drills and earthquake emergency plans.

**Keywords:** city-scale building performance assessment; human safety; elevator; earthquake; Monte Carlo

## 1. Introduction

Elevators are common nonstructural components of buildings. Owing to the rapid development of cities worldwide, elevators are being widely used in medium- and high-rise buildings [1]. According to the statistical data of the Beijing Municipal Administration of Market Supervision, 242,457 elevators had been constructed in Beijing by the end of 2019.

Earthquakes are among the most devastating natural disasters. The field investigation data of multiple earthquakes show that the elevator is one of the nonstructural components with high damage risk [2–7]. For example, the 1994 Northridge earthquake in California, USA caused damage to 968 traction elevators [3], and the 2008 Wenchuan earthquake in Sichuan, China caused damage to 1008 elevator systems [5]. Seismic damage to elevators is a significant threat to passenger safety as passengers may become trapped in elevators after an earthquake. For instance, after the 2018 Osaka earthquake in Japan, 339 people were trapped in elevators [6]. An accurate prediction of the number of passengers trapped in elevators (PTEs) in urban areas can provide support for developing targeted rescue plans to promote emergency efficiency by eliminating adverse aspects such as incomplete knowledge of the demand for rescue forces, improper allocation of

search personnel, and inefficient rescue operations during the valuable time of 72 h after an earthquake.

However, to the best of our knowledge, though PTE risk maps of urban areas are highly important for protecting the lives of people under earthquakes, studies regarding this topic are currently limited. In contrast, existing studies mainly focus on the seismic performance of elevators [8–12]. For example, Porter performed a regression analysis of post-earthquake survey data to obtain the fragility curves of hydraulic and traction elevators [8,9]. Wang et al. conducted a shaking table test to evaluate the seismic performance of elevators in five-story reinforced concrete buildings with and without seismic isolation [10]. Nguyen et al. investigated the vibration of the main and compensation ropes of an elevator system under earthquake excitation, considering the effect of varying rope lengths during the running of the elevator [11]. Han et al. established a finite element model of a counterweight elevator system in a 46-story structure and used the incremental dynamic analysis method to analyze its seismic fragility [12].

One step beyond the investigation of elevator seismic performance is to evaluate the earthquake-induced PTE risk for building complexes at a city scale, which is rarely discussed in existing studies given the complexity of the problem. Specifically, the city-scale assessment of PTE risk from earthquakes is accompanied by two challenges: (1) the significant number of buildings in an urban area renders it difficult to predict elevator damages at the city scale; (2) PTE risk evaluation based on elevator damage results is influenced by multiple factors with significant uncertainties, resulting in a lack of corresponding methods to quantify the city-scale PTE risk level.

Regarding the first challenge, considering the complexity of elevator damage mechanisms, an accurate prediction of structural seismic response plays a key role in predicting elevator damage. The finite element method has been widely used to predict the seismic performance of elevators in an individual building [12,13]. Nevertheless, this method cannot be applied at the city scale due to the significant amount of manpower and computing resources required to perform detailed finite element analysis of all elevator-equipped buildings in an urban area. The capacity spectrum approach and city-scale time history analysis (THA) can address the limitations of the finite element method. Hence, both methods have been widely applied in city-scale seismic simulations [14,15]. Although the capacity spectrum approach is more convenient to perform than city-scale THA, it is generally applied to low and multistory buildings whose deformation is dominated by the first vibration mode. However, elevators are generally implemented in high-rise buildings, in which the influence of higher-order modes cannot be neglected. For high-rise buildings, the capacity spectrum approach cannot rationally consider the characteristics of high-order vibration modes and ground motions [16]. By contrast, city-scale THA developed by the authors can be performed to conduct nonlinear THA for all buildings simultaneously in the target area based on structural dynamics theory, where the dynamic features of both structures and ground motions are considered comprehensively [17]. Therefore, city-scale THA was adopted in this study to address the first challenge.

Regarding the second challenge, the PTE risk under an earthquake is strongly connected to the elevator traffic level. In a specific urban area, elevator traffic is characterized by both spatial and temporal heterogeneity [18]. Specifically, spatial heterogeneity implies that the number of elevator passengers (NEP) in buildings with different occupancy types and different heights will exhibit different statistical characteristics, as will the number of passengers on different floors of the same building; temporal heterogeneity implies that the elevator traffic varies significantly at different hours of a day owing to the volatile pattern of human activities in each building. This sort of spatiotemporal heterogeneity means that the NEP in buildings during the earthquake will exhibit significant uncertainty. Consequently, the level of passenger entrapment risk will be significantly affected by the spatiotemporal heterogeneity of elevator traffic. However, obtaining the accurate number of passengers in each elevator of a building complex is difficult. Consequently, using a deterministic method to predict the number of PTEs is impractical. The Monte

Carlo method models the stochastic properties of a system by numerous random numerical experiments and thus can give approximate solutions to the abovementioned problem from a probabilistic perspective [19]. In this study, based on the field measurements of elevator traffic for several representative buildings, the Monte Carlo method was applied to address the second challenge.

To sum up, this study proposed a probability-based city-scale method to assess the earthquake-induced PTE risk in a building complex via the city-scale THA approach and Monte Carlo simulation to address the above challenges. The proposed method fills a gap in the research on earthquake-induced PTE risk at the city scale and provides a practical workflow with acceptable labor and time costs. Section 2 briefly describes the workflow, while Section 3 introduces the corresponding methodologies in detail. The case study and discussions are presented in Sections 4 and 5, respectively. Section 6 provides a summary of the findings.

## 2. Workflow

Figure 1 summarizes the workflow of the proposed method. The whole process is based on the Monte Carlo method. Each Monte Carlo simulation (referred to hereafter as a "realization") is assumed to have the same probability of occurrence. For each realization, all uncertain parameters are sampled (i.e., a single random value is selected from the corresponding distribution). In Figure 1, $s$ represents the number of realizations, and $I$ is the number of buildings in the target area. A minimum number of 1000 realizations are required (see Section 3.4). Specifically, the PTE assessment is the core part of the workflow and consists of three steps (Figure 1):

Step 1: Probability-based city-scale THA

Compared with the finite element method, the city-scale THA method developed by the authors previous studies can predict the seismic performance of a building complex more effectively. This method has been used successfully to simulate seismic damage in many cities such as San Francisco and Beijing [17]. Probability-based city-scale THA is performed in Step 1 to obtain engineering demand parameters (EDPs), such as the floor acceleration response of buildings under an earthquake. Specifically, the structural dynamic model of each building in the target area is automatically generated based on the building inventory data (i.e., structure type, number of stories, construction year, height, occupancy, and plane area). The building inventory data can be easily obtained from as-built drawings, city information models, and open-source geographical information system databases such as OpenStreetMap. Subsequently, the probability density functions (PDFs) of the mechanical parameters of each structural model are identified, and parameter values are stochastically sampled during each realization. Finally, city-scale THA is performed to predict the story-level dynamic responses of all buildings in the target area for a specific earthquake scenario. Note that the city-scale THA method can be used to predict the EDP of both drift- and acceleration-sensitive components. This work is limited to the prediction of story acceleration since elevators are assumed to be acceleration-sensitive components.

Step 2: Probability-based city-scale elevator damage assessment

This step is performed to obtain the seismic damage of each elevator in a building complex. Specifically, the story position of each elevator when the earthquake occurs is stochastically sampled in each realization based on the elevator inventory data (i.e., the number of elevators in each building as well as the elevator type and year of installation, which can be obtained using the method described in Section 3.2.1). Subsequently, based on the EDPs of the building complex simulated in Step 1 and elevator seismic fragility data, the damage probability of each elevator is identified. Consequently, the damage state of elevators can be randomly sampled during each realization.

Step 3: Probability-based city-scale elevator PTE assessment

This step is used to predict the number of earthquake-induced PTEs for a specific earthquake scenario. Based on the elevator traffic statistics (i.e., the time-varying mean and standard deviation values of NEP, which can be obtained as described in Section 3.3.1) and the elevator position data determined in Step 2, the number of passengers in each elevator at the time of earthquake occurrence can be sampled for each realization. Subsequently, the number of PTEs in each building is predicted based on elevator damage states obtained in Step 2.

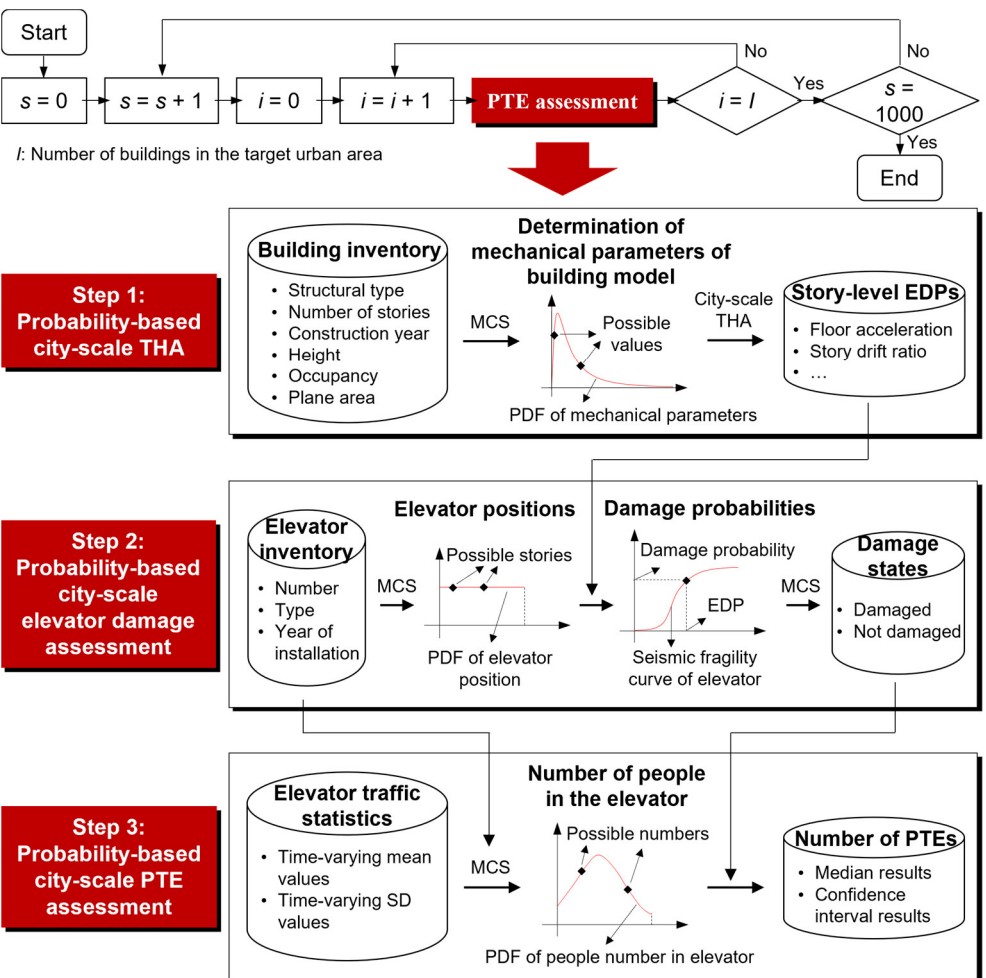

THA: time history analysis; MCS: Monte Carlo simulation; PDF: probability density function; EDP: engineering demand parameter; PTE: passenger trapped in elevator; SD: standard deviation

**Figure 1.** Workflow of the proposed method.

The underlying methodologies used in the abovementioned steps will be introduced in Section 3 in detail.

## 3. Methodologies

### 3.1. Probability-Based City-Scale THA

City-scale THA uses the multiple-degree-of-freedom (MDOF) shear model (Figure 2a) to simulate the structure whose deformation mode is dominated by shear deformation (e.g., the multistory frame structure), while the MDOF flexural–shear model (Figure 2b) is adopted for the structure with a large height-to-width ratio and a flexural–shear coupling deformation mode (e.g., high-rise frame–shear wall structure). The MDOF shear model assumes that the mass of the structure is concentrated at the story level and hence each story can be simplified as a mass point. The mass points between adjacent stories are

connected by a shear spring. By contrast, the MDOF flexural–shear model uses both a flexural spring and a shear spring to connect two adjacent stories, so that the high-order vibration mode of high-rise buildings can be captured. Specifically, the MDOF shear model can be used for popularly constructed multi-story structures, including multi-story steel structures, reinforced concrete frames, reinforced masonry structures, and unreinforced masonry structures; meanwhile, the MDOF flexural–shear model can be used for tall buildings (i.e., buildings taller than 10 stories) equipped with shear walls or braces. The authors verified the reliability of this method by comparing simulation results with field measurements of actual earthquakes [17]. The story mass and mechanical parameters of inter-story springs can be determined according to the design drawings. In the absence of design drawings, the authors proposed an automated model generation algorithm to estimate the parameters based on the building inventory data [17]. Note that the proposed parameter determination method, based on a simulated design procedure and the statistics obtained from the extensive experimental and analytical studies, can be easily applied to different regions with different design requirements.

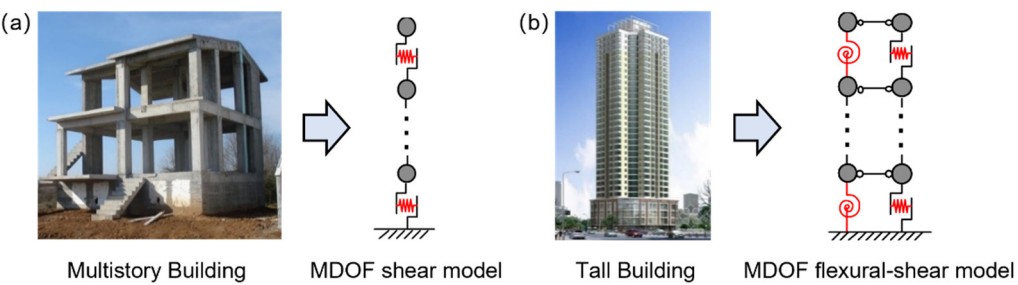

**Figure 2.** (**a**) MDOF shear and (**b**) flexural–shear models; MDOF: multiple-degree-of-freedom.

For brevity, given that taller buildings are more likely to be equipped with elevators, a reinforced concrete high-rise building is herein selected as an example to demonstrate the implementation of probability-based city-scale THA. Specifically, four steps are required [17]:

(1) In each realization, the first two order periods of the MDOF flexural–shear model are calculated based on the building height using empirical formulas. Then, the bending-to-shear stiffness ratio can be obtained. Consequently, the structure's elastic bending and shear stiffness values can be calculated, respectively.

(2) The design capacity of the shear and bending springs on each story is obtained by the mode-superposition response spectrum method. Subsequently, the distribution features of the yield and peak overstrength parameters, namely $\Omega_y$ and $\Omega_P$, respectively, are determined, based on which the $\Omega_y$ and $\Omega_P$ values are sampled stochastically in each realization. Consequently, the yield and peak points of the force–displacement skeleton curves of the shear and bending springs can be identified.

(3) The hysteresis parameters of the structure are determined.

(4) A nonlinear THA of the MDOF flexural-shear model is performed using the central difference method to obtain the seismic response of the structure.

In the abovementioned steps, $\Omega_y$ and $\Omega_P$ are two mechanical parameters with uncertainties. For a reinforced concrete high-rise building, both $(\Omega_y - 1)$ and $(\Omega_P - 1)$ follow a lognormal distribution, and their logarithmic mean values can be calculated using Equations (1) and (2), respectively [17]:

$$\ln(\Omega_y - 1) = 1.1941 - 0.2678DI \tag{1}$$

$$\ln(\Omega_p - 1) = 2.0252 - 0.2719DI \tag{2}$$

where *DI* is the fortification intensity, an indicator describing the seismic risk level of the site where the building is located. There are four options for seismic fortification intensity

according to the Chinese standard [20], i.e., 6, 7, 7.5, 8, 8.5, and 9, corresponding to the peak ground acceleration of 1.25, 2.20, 3.10, 4.00, 5.10, and 6.20 m/s² under the maximum considered earthquake, respectively. In each realization, the values of $\Omega_y$ and $\Omega_P$ sampled stochastically based on the above lognormal distribution can be used to establish the structural model and perform the THA to predict the story-level EDPs for a specific earthquake scenario.

### 3.2. Probability-Based City-Scale Elevator Damage Assessment

### 3.2.1. Elevator Inventory

Elevator inventory data include the number of elevators in each building, the elevator type, and the installation year. At the building scale, these data can be obtained via field investigation or as-built drawings with acceptable labor and time costs. In contrast, it is difficult to obtain city-scale elevator inventory data given that a real urban area consists of a significant number of buildings with various conditions. In general, among elevator inventory data, the number of elevators at the city scale is easier to obtain than the elevator type and installation year. Therefore, the following estimation method is suggested if the elevator type and installation year cannot be obtained via field investigation or relevant drawings.

Generally, the elevator type has two options: hydraulic and traction elevators. The US Federal Emergency Management Agency (FEMA) [21] defined four possible candidates for elevators using the year 1976 as the separator. Each candidate was assigned a Fragility ID, namely D1014.011, D1014.012, D1014.021, or D1014.022, representing traction elevators installed in 1976 and after, traction elevators installed pre-1976, hydraulic elevators installed in 1976 and after, and hydraulic elevators installed pre-1976, respectively. If the elevator type and installation year are unavailable, the method shown in Figure 3 is used to randomly select the elevator type based on the building inventory data for each realization. Specifically, for buildings constructed before 1976, four elevator types are possible, with each presenting a probability of 25%. For buildings constructed in 1976 or later, only two types of elevators are possible, namely D1014.011 and D1014.021, with each presenting a probability of 50%.

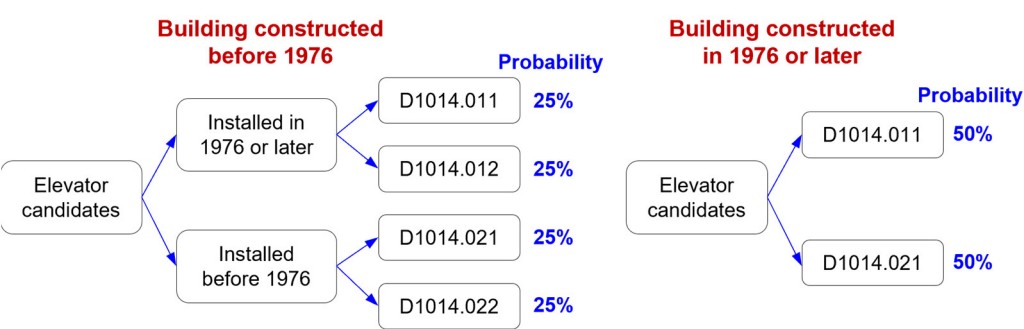

**Figure 3.** Method to determine elevator type and installation year when data are unavailable.

Note that the method shown in Figure 3 is used to determine elevator information only when the corresponding field investigation data and as-built drawings are unavailable. A field investigation is always recommended whenever possible. The simplified approach shown in Figure 3 can be used only when the time and labor costs of field investigation are unaffordable. Note also that an accurate prediction of the probability for each elevator candidate calls for large numbers of elevator survey data from various countries around the world, which are absent at present. The collaboration of related researchers worldwide is crucial to establish the database. Provided that the elevator datasets from different countries are established, the specific value of the probability for each elevator candidate can be correspondingly modified. At present, each candidate is assigned an identical probability value in the simplified method in Figure 3.

### 3.2.2. Elevator Position

In each realization, the story position of each elevator during an earthquake is determined based on a uniform distribution with a lower limit of 1 and the upper limit being the number of stories in the building. Mathematically, the probability $P_{i,j,k}$ that the *j*-th elevator of building *i* is at the *k*-th story when the earthquake occurs is expressed as follows:

$$P_{i,j,k} = \frac{1}{K_i} \tag{3}$$

where $K_i$ is the number of stories in building *i*.

### 3.2.3. Damage Probability and Damage State

The damage state of elevators can be determined based on the seismic fragility curve that describes the relationship between damage probability and elevator EDP. The elevator EDP is calculated according to the story-level EDP predicted in Section 3.1 and the elevator position obtained in Section 3.2.2. The seismic fragility curve of an elevator can be obtained from the seismic fragility database of building components established by the FEMA [21] or from the literature [8,9].

For brevity, the elevator candidate D1014.011 is used here as an example to illustrate the elevator damage state determination method based on the seismic fragility data from FEMA. There is only one damage state for elevators in the FEMA database. The failure probability $F(EDP_{i,j})$ of D1014.011 follows a lognormal distribution with a median $\theta_{i,j}$ of 3.9 m/s² and a logarithmic standard deviation $\beta_{i,j}$ of 0.45, as expressed in Equation (4) [21,22]:

$$F\left(EDP_{i,j}\right) = \Phi\left(\frac{\ln\left(EDP_{i,j}/\theta_{i,j}\right)}{\beta_{i,j}}\right) \tag{4}$$

where $EDP_{i,j}$ is the peak acceleration of the *j*-th elevator of building *i* during an earthquake, and $\Phi(\cdot)$ is the cumulative distribution function (CDF) of standard normal distribution. Consequently, the damage state of the *j*-th elevator of building *i*, namely $DS_{i,j}$, follows Bernoulli distribution, i.e., $DS_{i,j} \sim B(1, F(EDP_{i,j}))$, where $DS_{i,j} = 1$ implies that the elevator is damaged while $DS_{i,j} = 0$ implies that the elevator can still be used.

### 3.3. Probability-Based City-Scale PTE Assessment

### 3.3.1. Elevator Traffic Statistics

As stated in Section 1, elevator traffic exhibits complex spatiotemporal characteristics. As a result, it is impractical to accurately obtain the spatiotemporal data of elevator traffic for each building in an urban area given huge labor and time costs. To address this problem, a practical method for estimating the spatiotemporal distribution feature of elevator traffic for a building complex is proposed, as shown in Figure 4. In the absence of detailed field investigation data of elevator traffic for each story of a building, the proposed estimation method assumes that the statistical values of elevator passengers at different stories exhibit a linear relationship. In realistic situations, the specific occupancy of each story affects the elevator traffic level at that story, rendering the relationship between the NEP and the story extremely complex. However, obtaining these data at a city scale is difficult. The proposed method provides a feasible strategy for the city-scale estimation of NEP in building complexes. It consists of three steps:

Step 1: Typical building selection

Several typical buildings in the target area are first filtered out according to building occupancy and height, given the fact that the occupancy and height of a building are the main factors affecting the elevator traffic level. Specifically, all elevator-equipped buildings in the target area should be categorized into several categories based on building occupancy and height. In this study, a height of 10 stories is suggested as the threshold to

categorize buildings as a field investigation performed by the authors indicates that buildings with fewer than 10 stories and those with 10 stories or more exhibit significant differences in the elevator traffic statistics. Furthermore, at least two typical buildings need to be selected for each category to ensure that the elevator traffic observations in the following Step 2 are representative.

Step 2: Estimation of ground-level elevator traffic statistics

First, the ground-level elevator traffic statistics of the typical buildings are observed and counted on-site. In addition, videos of the elevator entrance can be used in conjunction with target tracking technology (e.g., YOLOv5 at https://github.com/ultralytics/yolov5 (2 January 2023)) to automatically calculate the NEP. The NEP on the ground story of a typical building, $ET_{\text{ground,typical}}$, is expressed as

$$ET_{\text{ground,typical}} = \frac{TP_{\text{in}} + TP_{\text{out}}}{RT} \tag{5}$$

where $TP_{\text{in}}$ and $TP_{\text{out}}$ represent the total number of passengers entering and exiting the elevator during a certain observation period, respectively; and $RT$ is the total number of times to record $TP_{\text{in}}$ and $TP_{\text{out}}$ values, correspondingly.

Subsequently, the ground-level NEP of a target building belonging to the same category as the typical building, $ET_{\text{ground,common}}$, can be calculated as

$$ET_{ground,common} = \frac{ET_{ground,typical} \cdot BA_{common} \cdot NoE_{typical}}{BA_{typical} \cdot NoE_{common}} \tag{6}$$

where $BA_{\text{common}}$ and $BA_{\text{typical}}$ are the floorages of the target and typical buildings, respectively; and $NoE_{\text{common}}$ and $NoE_{\text{typical}}$ are the numbers of elevators in the target and typical buildings, respectively.

Based on continuous observation in a typical building for a whole day, a time-varying estimate of the ground-level elevator traffic in a building complex can be obtained. By repeating these observations for several days, multiple time-varying estimates can be obtained and used to calculate the time-varying mean and standard deviation of the ground-level elevator traffic in each building.

Step 3: Estimation of elevator traffic statistics on all stories

Based on the ground-level traffic statistics, the NEP of the remaining stories can be estimated using Equations (7) and (8):

$$ET_{i,j,k} = \frac{K_i + 1 - k}{K_i} \cdot ET_{i,j,1} \tag{7}$$

$$\sigma_{i,j,k} = \frac{K_i + 1 - k}{K_i} \cdot \sigma_{i,j,1} \tag{8}$$

where $ET_{i,j,k}$ and $\sigma_{i,j,k}$ are the mean and standard deviation of passengers of the $j$-th elevator at the $k$-th story of building $i$, respectively; and $ET_{i,j,1}$ and $\sigma_{i,j,1}$ are the mean and standard deviation on the ground story, respectively.

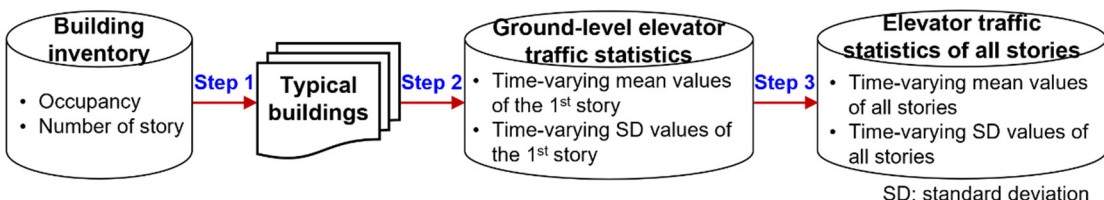

**Figure 4.** Method to estimate spatiotemporal data of elevator traffic for building complexes.

3.3.2. Number of Elevator Passengers

The NEP can be sampled randomly in realizations by assuming that it follows a truncated Gaussian distribution with the PDF expressed as

$$f\left(x; ET_{i,j,k}, \sigma_{i,j,k}, a, b\right) = \frac{\phi((x - ET_{i,j,k})/\sigma_{i,j,k})/\sigma_{i,j,k}}{\Phi\left((b - ET_{i,j,k})/\sigma_{i,j,k}\right) - \Phi((a - ET_{i,j,k})/\sigma_{i,j,k})} \tag{9}$$

where $x$ is the number of passengers in an elevator; $\phi(\cdot)$ and $\Phi(\cdot)$ are the PDF and CDF of the standard normal distribution, respectively; $a$ is the lower limit of the truncated Gaussian distribution, which is equal to zero; and $b$ is the upper limit of the distribution, which is equal to the maximum number of passengers that can be carried by the elevator. The maximum number of passengers can be determined based on building occupancy when there are no field investigation data. Specifically, based on a field investigation performed by the authors for a real urban area, the maximum number of elevator passengers for residential buildings can be randomly taken from 4–8 in each realization; that for hospital, educational, and commercial buildings can be stochastically sampled from 8–12; while that for industrial buildings can be randomly taken from 8–16. Nevertheless, a field investigation is always recommended whenever possible.

To sum up, the method in Section 3.2.2 is used to determine the story position $k$ of each elevator when an earthquake occurs. Subsequently, the method in Section 3.3.1 is used to identify the $ET_{i,j,k}$ and $\sigma_{i,j,k}$ values. Finally, the NEP can be sampled stochastically based on Equation (9).

### 3.3.3. Number of PTEs

If the damage state of an elevator equals 1 (see Section 3.2.3), the passengers inside it are identified as the PTE. The number of PTEs of building $i$ in the $s$-th realization, $PTE_{s,i}$, is expressed as

$$PTE_{s,i} = \sum_{j=1}^{J_i} PTE_{s,i,j} \tag{10}$$

where $PTE_{s,i,j}$ is the PTE number of the $j$-th elevator in building $i$ in the $s$-th realization; and $J_i$ is the number of elevators in building $i$. Furthermore, the total number of PTEs in the target area in the $s$-th realization, $PTE_s$, can be calculated as

$$PTE_s = \sum_{i=1}^{I} PTE_{s,i} \tag{11}$$

where $I$ is the number of buildings in the target area. Finally, by repeating a sufficient number of realizations, the median and confidence interval of PTE can be predicted based on the $PTE_{s,i}$ and $PTE_s$ values of all realizations.

### 3.4. Number of Realizations

To ensure the accuracy of the Monte Carlo method, a sufficient number of realizations is required. A corresponding numerical experiment indicated that the proposed method could yield the desired accuracy after 1000 realizations. For brevity, the results of the case with the highest degree of variation in Section 4 are presented in Figure 5, illustrating the changes in the normalized median and standard deviation of the PTE for both the individual building and the whole campus with respect to the number of realizations. It can be seen that accurate results can be obtained after 1000 realizations. Hence, the subsequent study adopted 1000 as the number of realizations.

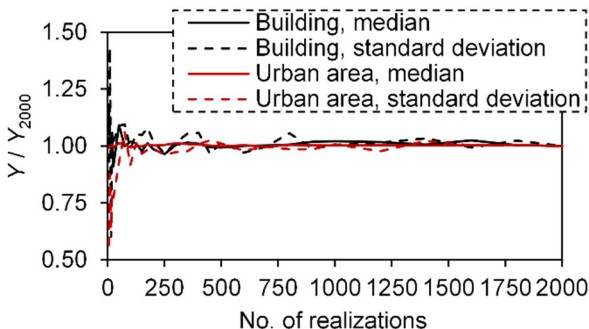

**Figure 5.** Determination of the number of realizations.

## 4. Case Study

### 4.1. Study Area

A real-world urban area, namely the Tsinghua University campus, was selected as the case study to demonstrate the feasibility of the proposed method. The campus, located in northwestern Beijing, China, features an area of approximately 4 km². The seismic fortification intensity at this site is eight [20]. A field investigation was conducted to obtain the building inventory data of 619 buildings. The occupancy type and number of stories of the buildings are shown in Figure 6. The campus is dominated by residential and office buildings, with high-rise buildings mainly located in the northeast and southeast regions.

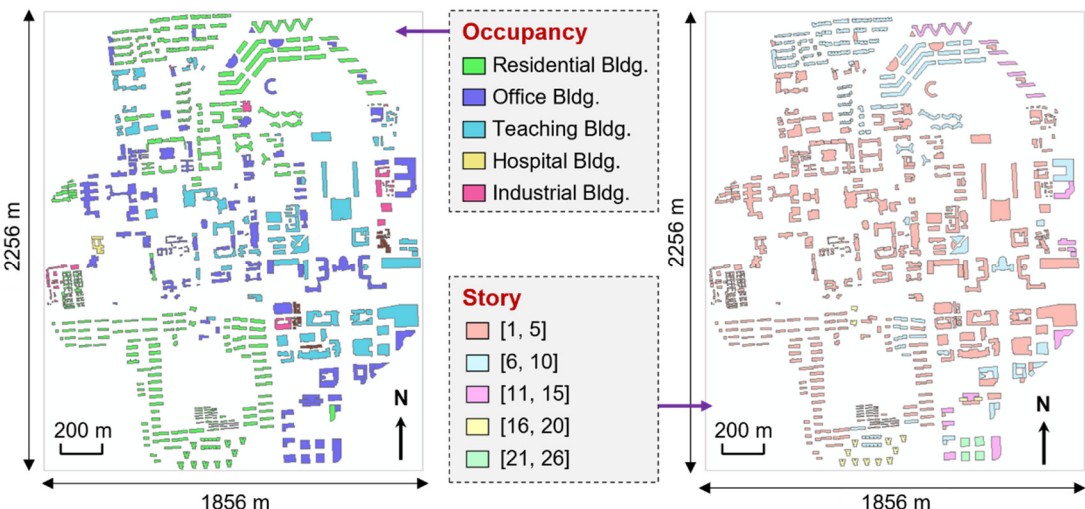

**Figure 6.** Occupancy and story of buildings on the Tsinghua University campus.

The proposed method can be used for two types of assessments, i.e., scenario- and intensity-based assessments. The first type of assessment can be used for specific earthquake scenarios (e.g., historical earthquake events in the target area), while the second is applied for specific earthquake intensities (e.g., the design basis earthquake, DBE). The scenario- and intensity-based assessment results of the study area will be introduced in Sections 4.4 and 4.5, respectively.

### 4.2. Elevator Inventory Data

For the Tsinghua University campus, a comprehensive field investigation was performed to obtain the elevator inventory data. The number of elevators in each building is shown in Figure 7. A total of 251 elevators were installed in 82 buildings, with traction elevator the predominant elevator type. Most of the elevators were installed in the year 2000 or later. The FEMA database was adopted to identify elevator seismic fragility in the

study area given that elevators, as a common nonstructural component in modern buildings, do not exhibit significant differences between different countries.

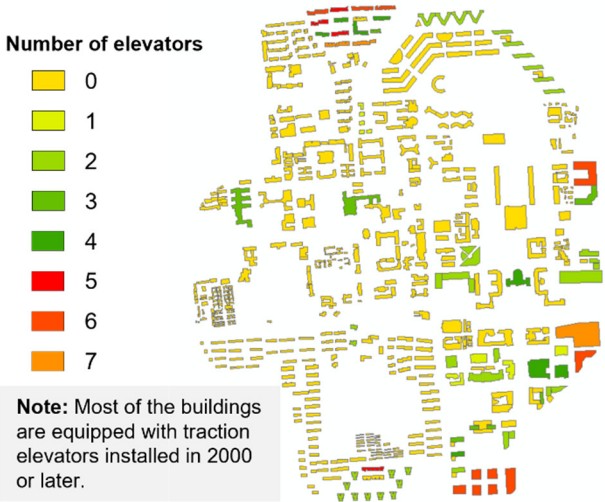

**Figure 7.** Number of elevators in buildings of the Tsinghua University campus.

*4.3. Elevator Traffic Data*

Elevator-equipped buildings on the Tsinghua University campus are categorized into residential, office, and teaching buildings, whereas buildings with other occupancies are low-rise buildings without elevators. Considering the significant difference between the elevator traffic of student dormitories and ordinary residences, the residential buildings on the campus were further classified into two categories: student dormitories and ordinary residences. Consequently, the typical buildings of the campus were selected from four occupancy categories, i.e., office buildings, student dormitories, ordinary residences, and teaching buildings. Furthermore, the typical buildings in each occupancy type were subcategorized into two categories based on the number of stories, i.e., buildings lower than 10 stories (denoted as L building hereinafter) and buildings with 10 stories or more (denoted as H buildings hereinafter). As a result, the campus buildings were classified into eight categories, and two typical buildings were selected from each category, as shown in Table 1.

**Table 1.** Typical buildings on the Tsinghua University campus.

| Buildings Lower than 10 Stories | | | Buildings with 10 Stories or More | | |
|---|---|---|---|---|---|
| Name | Occupancy | Stories | Name | Occupancy | Stories |
| Mengminwei S&T Bldg. | Office building | 8 | Innovation Mansion | Office building | 13 |
| Medical Science Bldg. | Office building | 3 | SP Bldg. B | Office building | 26 |
| Bldg. 29 | Student dormitory | 6 | Zijing Student Apt. No.14 | Student dormitory | 15 |
| Bldg. 21 | Student dormitory | 6 | Zijing Student Apt. No.23 | Student dormitory | 12 |
| Heqingyuan No.12 | Ordinary residence | 9 | Tall Bldg. No.1 | Ordinary residence | 17 |
| Heqingyuan No.1 | Ordinary residence | 7 | Lanqiying No.8 | Ordinary residence | 20 |
| 6th Teaching Bldg. Zone A | Teaching building | 6 | 6th Teaching Bldg. Zone B | Teaching building | 10 |
| Precision Instruments Dept. | Teaching building | 4 | Liuqing Bldg. | Teaching building | 11 |

The $RT$, $TP_{in}$, and $TP_{out}$ of the typical buildings were monitored every 10 min, and a piecewise linear function was applied to fit the statistical data. After repeated observations for 7 days, the time-varying mean and standard deviation of the ground-level NEP of each elevator for the typical buildings were calculated (Figure 8). It is noteworthy that data in Figure 8 were obtained during workdays (i.e., from Monday to Friday, excluding public holidays). Nevertheless, the same method can be used for weekends and holidays.

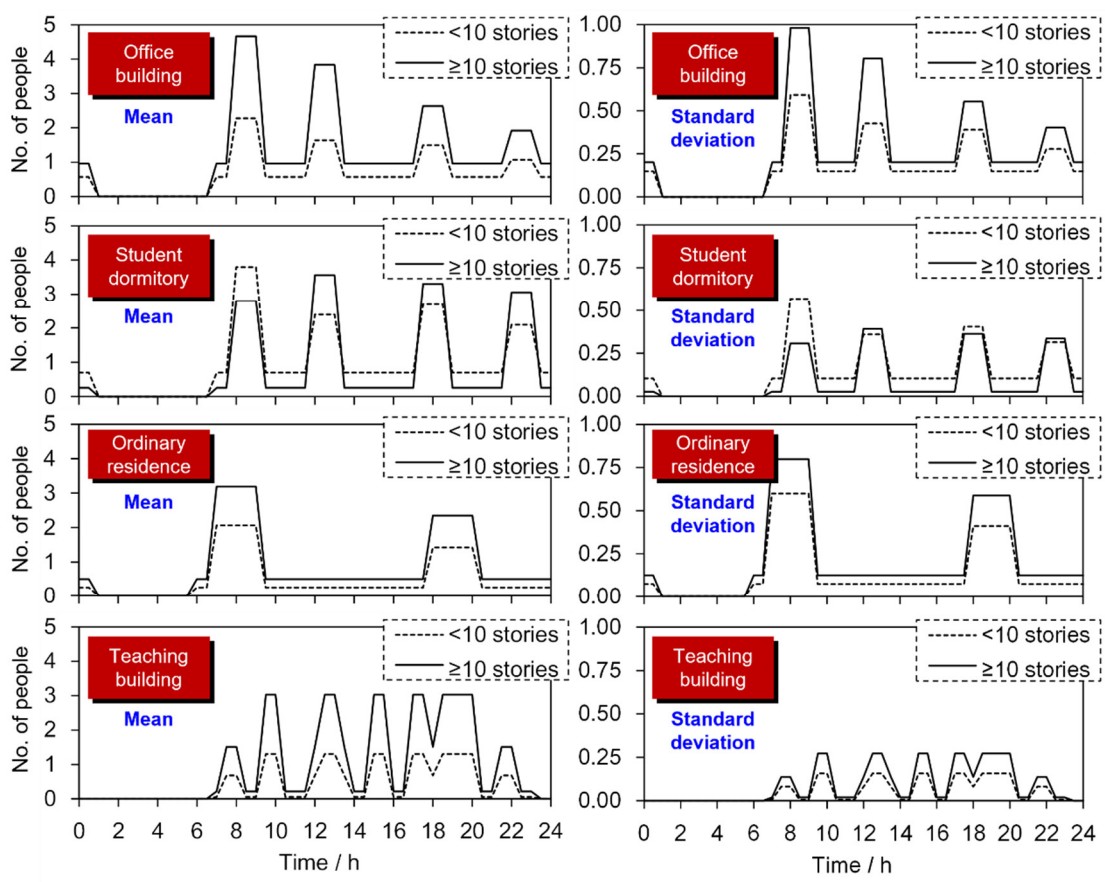

**Figure 8.** Time-varying traffic statistics for each elevator per 1000 m² floorage of typical buildings.

It can be seen from Figure 8 that there are significant differences in the peak periods of elevator traffic for buildings with different occupancies. The peak periods for office buildings and student dormitories are similar in that there are four peak periods, i.e., morning, midday, evening, and late-night peaks. For office buildings, the morning peak is the highest while the late-night peak is the lowest. Moreover, because employees working in low-rise buildings prefer to use stairs instead of waiting for an elevator, the elevator passengers of H buildings are significantly more than those of L buildings. For student dormitories, the L buildings exhibit the highest elevator traffic level at the morning peak, whereas the H buildings exhibit the highest elevator traffic level at the midday peak, which is attributable to the fact that the L and H buildings house primarily undergraduate and postgraduate students, respectively. Compared with undergraduate students, postgraduate students are less constrained by courses; consequently, their schedules are more flexible, which results in greater uncertainties in their work and rest patterns. An interesting finding is that the postgraduate students were more likely to leave and return to their dormitories later in the day, as reflected by the fact that the midday, evening, and late-night peaks in H buildings are higher than the morning peak. The ordinary residences have morning and evening peaks. Owing to the university's class schedule, the teaching buildings have multiple peaks. Apart from the dormitories, the peaks of the H buildings generally exceed those of the L buildings. Moreover, the peak-time standard deviations

of the office and ordinary residences are greater than those of the dormitories and teaching buildings. This is because, unlike staff, students are generally preoccupied with class attendance and research, which translates to a more structured and less random activity pattern.

### 4.4. Scenario-Based Assessment

The 1679 Sanhe–Pinggu earthquake was selected as the target scenario given that it was the most recent strong earthquake in Beijing. The simulated ground motion [23] with a peak ground acceleration (PGA) of 5.6 m/s$^2$ obtained from the Institute of Geophysics of China Earthquake Administration was used as input (Figure 9). The same PGA was used for all buildings for simplicity as the campus area was not large.

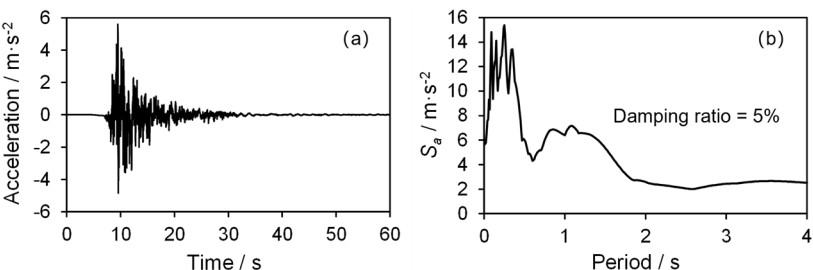

**Figure 9.** (**a**) Ground motion and (**b**) response spectrum of the 1679 Sanhe–Pinggu earthquake.

Figure 10 shows the number of PTEs on the campus when the earthquake occurs at different times on a weekday. The curve indicates four local peaks, which correspond to the time of earthquake (TOE) of 8:00, 12:30, 18:30, and 22:00, respectively. Considering that the PTE risk level is affected by multiple factors with significant uncertainties, Figure 10 presents the time-varying median values in conjunction with the 95% confidence intervals to provide government agencies with a sound reference. At the TOE of 8:00, the PTE number reached its maximum value, with a median of 238 and a 95% confidence interval from 201 to 276. The median values of the remaining three peaks are 188, 194, and 133, respectively. In addition, if the earthquake occurs at an off-peak time of elevator traffic, e.g., 16:30, the PTE number is approximately a quarter of the maximum value of 238.

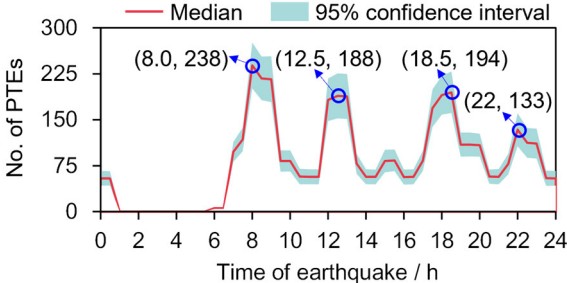

**Figure 10.** Number of PTEs on the campus when the 1679 Sanhe–Pinggu earthquake occurs at different times on a weekday.

Figure 11 shows the predicted number of PTEs in buildings at the abovementioned four local peak times. The buildings in the east and southeast regions of the campus have a higher risk of PTE, among which Buildings 1, 2, 3, and 4 indicate the highest risks. Buildings 3 and 4 are office buildings with a high traffic level per elevator, and these two buildings are equipped with six elevators. As a result, the number of PTEs is larger than 10 people. Meanwhile, Buildings 1 and 2 are five-story teaching buildings. Although the traffic level per elevator in the teaching buildings is relatively low, Building 1 comprises seven

elevators, which is the highest number of elevators among the campus buildings; therefore, its PTE risk is relatively high. It is noteworthy that the number of elevators in Building 2 is not large. The reason for the large number of PTEs in this building is that its first-order natural period of 0.38 s is within the peak range of the response spectrum of the ground motion (Figure 9). Consequently, a greater seismic response implies a higher risk of elevator damage. These results indicate that the proposed method can rationally consider the influences of multiple factors, such as the characteristics of both ground motions and structures, the number of elevators, and the spatiotemporal features of elevator traffic.

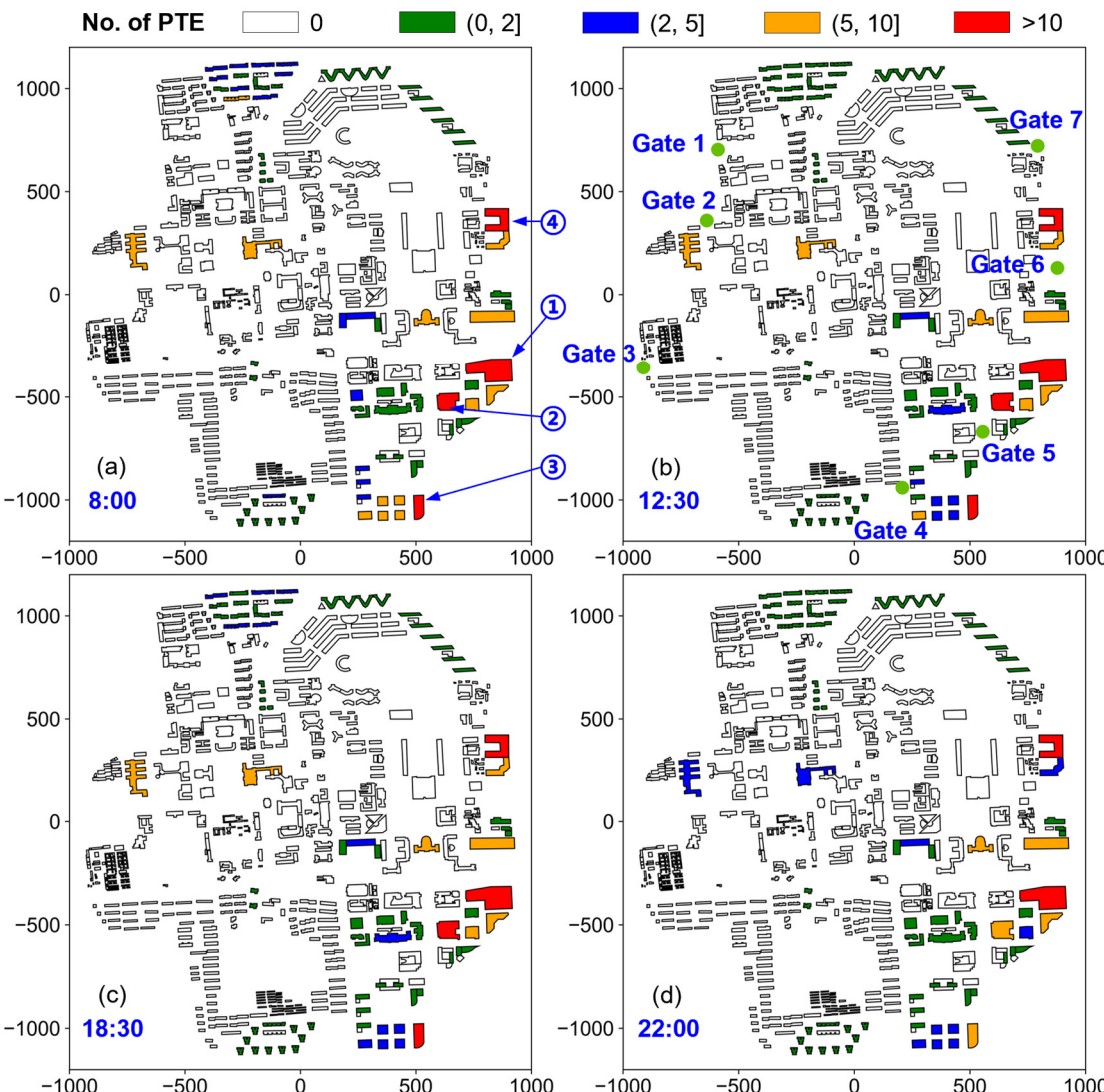

**Figure 11.** Number of PTEs in buildings when the 1679 Sanhe–Pinggu earthquake occurs at (**a**) 8:00, (**b**) 12:30, (**c**) 18:30, and (**d**) 22:00 on a weekday.

Furthermore, the campus has seven entrances that can be used by rescue forces in post-earthquake emergencies (Figure 11). The optimal entrance for emergency forces is determined based on the distance to reach the area with the largest number of trapped people. The analysis results indicate that Gate 5 is the optimal choice under the 1679 Sanhe–Pinggu earthquake. Note that a discussion of different entrance selection criteria is out of the scope of this study. Once the PTE number of each building is obtained through the proposed method, one can quantify the scores of different entrances based on any reasonable criterion.

*4.5. Intensity-Based Assessment*

Intensity-based assessment requires a set of ground motion records that reflect the site characteristic of the target area. Generally, at least eleven ground motion records are needed [21]. Therefore, eleven ground motion records (Table A1 in Appendix A) that matched the target response spectrum of the campus were identified from the NGA-West2 database [24] established by the US Pacific Earthquake Engineering Research Center. A target response spectrum was generated according to the design basis earthquake level specified in the Chinese standard [20]. The response spectra of the selected records and the corresponding averaged spectrum are shown in Figure 12, demonstrating that the selected records can reflect the site features of the campus. Furthermore, four earthquake intensities with the PGA of 0.1 g, 0.2 g (DBE level), 0.3 g, and 0.4 g (maximum considered earthquake (MCE) level) were considered to evaluate the PTE risk of the campus. For each intensity, the average of the simulated results under the eleven ground motions was identified as the final assessment result.

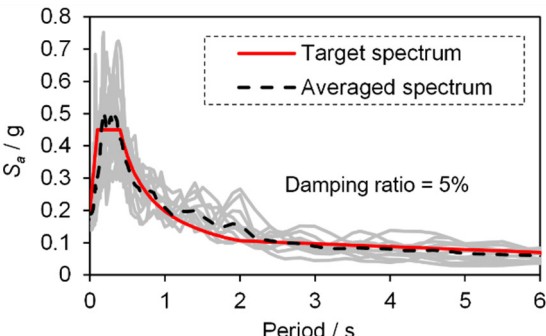

**Figure 12.** Response spectra of ground motions selected from the NGA-West2 database.

Figure 13 shows the number of PTEs on the campus under earthquakes of four different intensities. As the PGA increases, the mean value increases while the relative variation level decreases. Given a TOE of 8:00, the maximum number of DBE-induced PTEs is 41 while that of MCE-induced PTEs reaches 195. Consistent with the results under the 1679 Sanhe-Pinggu earthquake, the curves in Figure 13 exhibit four local peaks, corresponding to the TOEs of 8:00, 12:30, 18:30, and 22:00, respectively. The maximum number of PTEs is observed at the TOE of 8:00 because the elevator traffic level of the office building, student dormitory, and ordinary residence reaches its highest at the same time during a workday at 8:00. The numbers of PTEs at the TOEs of 12:30 and 18:30 are similar, and the local peaks at the TOE of 22:00 are significantly lower than the other peaks. The numbers of PTEs at an off-peak time of elevator traffic (e.g., 16:30) are approximately a quarter of those at the TOE of 8:00. It is noteworthy that the TOEs of the four local peaks overlap to a large extent with the peak hours of traffic in Beijing, thus potentially affecting the speed and efficiency of emergency responses. The results highlight the importance of formulating targeted emergency plans based on pre-earthquake assessments.

Figure 14 shows the number of PTEs in buildings when DBE- and MCE-level scenarios occur at 8:00 on a weekday. Twenty-eight buildings indicate the PTE risk under the DBE, while all elevator-equipped buildings have PTEs under the MCE. When the earthquake is of MCE intensity, ten buildings (i.e., nine office buildings and one teaching building) have more than five PTEs, whereas two buildings (i.e., Buildings 1 and 4 in Figure 11) indicate more than ten PTEs; the high-risk buildings are mainly located in the east and southeast regions of the campus. Based on the results, Gate 1 (Figure 11) is the optimal entrance for emergency forces to implement a rescue when the earthquake is at the DBE level, while Gate 5 (Figure 11) is the best choice under the MCE.

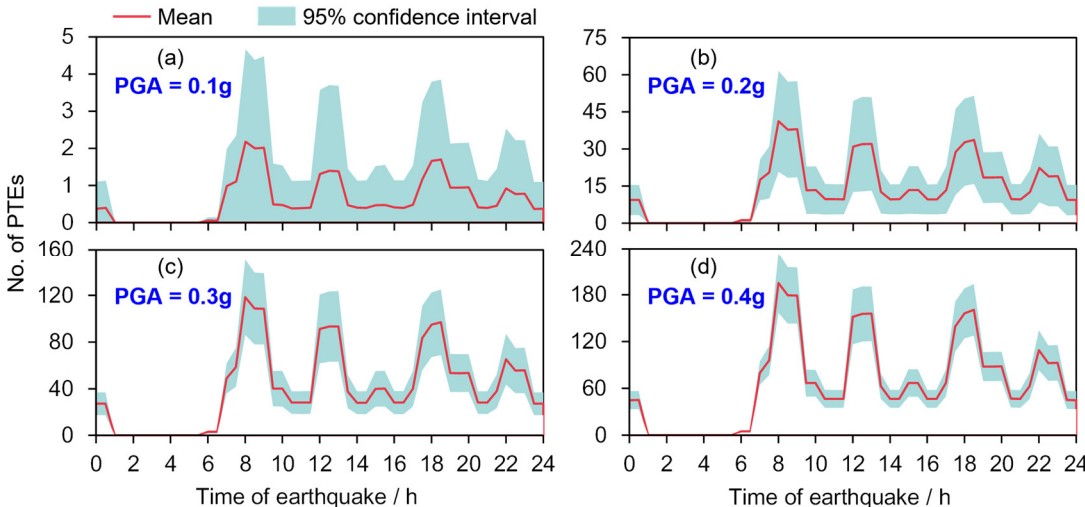

**Figure 13.** Number of PTEs on the Tsinghua University campus under earthquakes with different intensities (earthquakes occur on weekdays): (**a**) PGA = 0.1 g, (**b**) PGA = 0.2 g, (**c**) PGA = 0.3 g, and (**d**) PGA = 0.4 g.

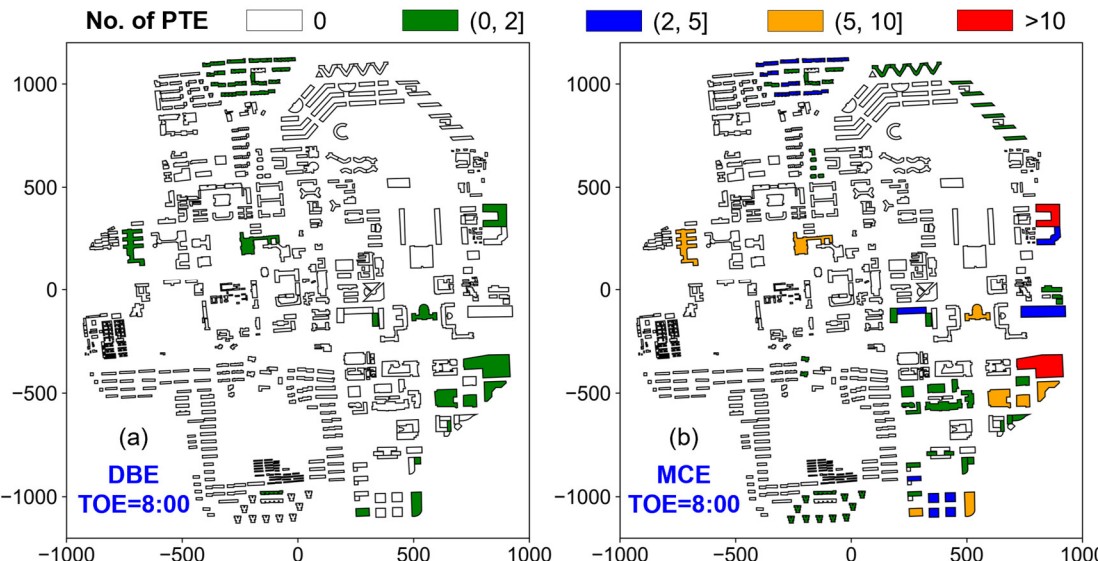

**Figure 14.** Number of PTEs in buildings when (**a**) DBE- and (**b**) MCE-level earthquakes occur at 8:00 on a weekday.

Furthermore, based on the simulations with the selected ground motions being adjusted to other PGA levels (e.g., 0.25 g and 0.35 g), the fragility curve of PTE risk on the campus could be obtained by fitting the relationship between the earthquake intensity and the number of PTEs. In this section, a total of 132 scenarios (i.e., 11 ground motions with 12 intensity levels) were simulated to generate fragility curves. The CDF of the lognormal distribution is widely used to fit fragility curves in the field of earthquake engineering [21]. Hence, it was also used in this study. As shown in Equation (12), the fitting function features three parameters, i.e., $\mu$, $\sigma$, and $k$. Figure 15 indicates the fitted fragility curves for the TOEs of 8:00, 12:30, 18:30, and 22:00, provided that the earthquake occurs on weekdays. For brevity, the fragility curves for other TOEs are not presented. At the TOEs of 8:00, 12:30, 18:30, and 22:00, the limit values of PTEs are 218, 179, 187, and 124, respectively. Furthermore, the maximum number of PTEs under the MCE is approximately five times that under the DBE.

$$F(x) = \frac{k}{\sqrt{2\pi}\sigma} \int_{-\infty}^{x} e^{\frac{-(t-\mu)^2}{2\sigma^2}} dt \tag{12}$$

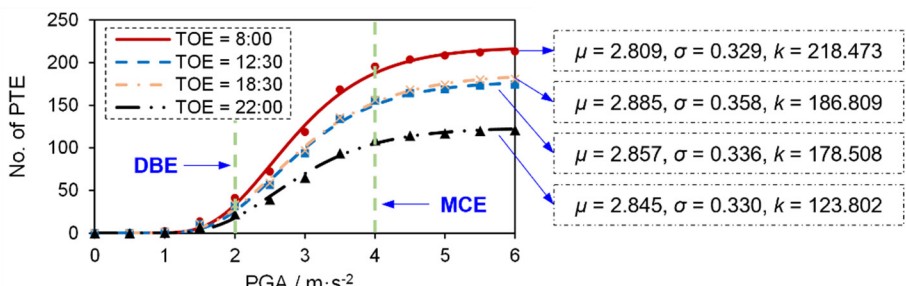

**Figure 15.** Fitted fragility curves when an earthquake occurs at 8:00, 12:30, 18:30, and 22:00 on weekdays.

## 5. Discussions

Although a lot of studies have been conducted to investigate the seismic performance of elevator systems, further discussions regarding the risk of passenger entrapment due to elevator seismic damages are rarely reported, particularly at the city scale. In this study, a novel probability-based city-scale method for assessing the number of earthquake-induced PTEs was proposed, in which city-scale THA was performed to simulate the seismic response of a building complex, and the Monte Carlo method was applied to generate probability-based assessment results considering the uncertainty of multiple factors (i.e., the mechanical properties of buildings and elevators, the elevator story position, and the spatiotemporal features of elevator traffic) affecting the PTE risk level. Furthermore, the proposed method can be used to perform both scenario- and intensity-based assessments of PTE risks, thereby providing references for government agencies to develop targeted emergency plans.

The most convenient method to estimate the PTE number of an urban area is to use an empirical formula or a neural network regressed based on historical disaster data. However, this sort of data-driven method strongly relies on the availability of corresponding historical data. As a result, this kind of method is generally only applicable to specific locations and earthquake scenarios, exhibiting poor universality. In contrast, the workflow proposed in this study is a physics-driven method that can be used in different regions and for different earthquake scenarios.

The PTE risk level of an urban area varies significantly by the TOE owing to the temporal nature of elevator traffic in building complexes. The number of PTEs is relatively large when an earthquake occurs at the peak period of elevator traffic. In the case study of the Tsinghua University campus, as shown in Figures 10 and 13, the maximum number of PTEs is observed at the TOE of 8:00–9:00, followed by at the TOE of 17:30–18:30; the number of PTEs at the TOE of 12:00–13:00 is slightly lower than that at the TOE of 17:30–18:30; and the number of PTEs at the TOE of 22:00–23:00 is the smallest among the above-mentioned four local peaks. When the earthquake occurs during the off-peak hours of elevator traffic (e.g., at 16:30), the number of PTEs is approximately a quarter of that at the TOE of 8:00.

Additionally, the PTE risk level differs significantly for different buildings in an urban area due to the spatial heterogeneity of elevator traffic in building complexes, thus affecting post-earthquake rescue implementations (e.g., the determination of the optimal entrance into the stricken area). In the case study of the Tsinghua University campus, as shown in Figures 11 and 14, the buildings in the southeast region indicate a higher risk of PTE. Moreover, office buildings indicate the highest risk of PTE, followed by teaching buildings. Although the elevator traffic level is generally higher in the H buildings than

in the L buildings (Figure 8), large numbers of PTEs are observed in both types of buildings given the influences of structural seismic response characteristics and the number of elevators. The results emphasize the complex relationship between the PTE risk, earthquake characteristics, and building and elevator inventory. Moreover, Gate 1 is identified as the optimal entrance for post-earthquake rescues under the DBE, while Gate 5 is the best choice when the earthquake reaches the MCE level.

Furthermore, the proposed method can be used to generate fragility curves depicting the quantitative relationship between the number of PTEs in an urban area and earthquake intensity. For the Tsinghua University campus, the maximum number of PTEs under the MCE is approximately five times that under the DBE (Figure 15). These fragility curves enable a comprehensive understanding of the PTE risk in the target area under different earthquake intensities, thus promoting targeted earthquake emergency preparations.

The proposed method has the potential to provide support for virtual rescue drills. Specifically, by combining the simulation results with a city information model [15] and virtual reality technologies [25], a high-fidelity post-earthquake virtual environment can be realized for the training of rescue forces such as firefighters and paramedics, thereby facilitating emergency efficiency.

This study presents the following limitations:

(a) For a specific urban area, a certain functional connection typically exists between different buildings, which implies that a correlation exists among the probability distributions of NEP in different buildings. It is currently challenging to identify this type of correlation at a city scale. Therefore, this study does not consider the impact of such correlation problems but assumes that the random variable of NEP is independent of each other for different buildings.

(b) In addition, a linear function was used to describe the distribution pattern of elevator passengers at different stories of the same building, which is an idealized assumption. Different stories of a building may exhibit different occupancies. For example, the bottom floors of a high-rise building may be used for commercial activities, whereas the remaining floors are for residential purposes. The spatial variations in building occupancy will significantly affect the distribution pattern of elevator passengers on different stories [26].

(c) Furthermore, the elevator traffic data of a specific building may be affected by the weather and season. Long-term observation of elevator traffic is crucial to the accurate prediction of city-scale PTE risks.

In subsequent research, the authors will continue improving elevator traffic data resolution. Nevertheless, the proposed method fills a gap in the research on earthquake-induced PTE risk at a city scale and provides a practical assessment workflow with acceptable labor and time costs.

## 6. Conclusions

A probability-based city-scale method for assessing the number of earthquake-induced PTEs in a building complex is proposed in this study. The feasibility of the method was demonstrated based on a case study of the Tsinghua University campus. Conclusions are presented as follows:

(1) In the proposed method, city-scale THA is performed to simulate the seismic response of a building complex, and the Monte Carlo method is applied to generate probability-based results by considering the uncertainty of multiple factors (i.e., the mechanical properties of buildings and elevators, the elevator story position, and the spatiotemporal characteristics of elevator traffic) affecting the PTE risk level.

(2) The proposed method can be used to perform both scenario- and intensity-based assessments of earthquake-induced PTE risks, thereby having the potential to provide

support for applications such as virtual rescue drills and earthquake emergency plans.

(3) The spatiotemporal nature of elevator traffic significantly affects the PTE risk in a building complex. In the case study of the Tsinghua University campus, the number of PTEs when an earthquake occurs during the off-peak hours of elevator traffic is approximately a quarter of that when the earthquake occurs at the morning peak time; the office building exhibits the highest PTE risk among the campus buildings; and the high-risk buildings are mainly located in the east and southeast regions of the campus under the MCE.

(4) The maximum number of PTEs on the Tsinghua University campus under the MCE reaches 195, approximately five times that under the DBE. Gate 1 is identified as the optimal entrance for post-earthquake rescues under the DBE, while Gate 5 is the best choice under the MCE.

(5) The fragility curves depicting the quantitative relationship between the number of PTEs in an urban area and the earthquake intensity enable a comprehensive understanding of the PTE risk in the target area under different earthquake intensities, thus promoting targeted earthquake emergency preparations.

(6) The proposed method fills a gap in the research on earthquake-induced PTE risk at a city scale and provides a practical assessment workflow with acceptable labor and time costs. Additionally, compared to the data-driven solution, the workflow is a physics-driven method that has no geographical limitations.

**Author Contributions:** Conceptualization, D.G. and X.L.; Methodology, D.G.; Software, D.G.; Formal Analysis, D.G.; Investigation, D.G. and Y.W.; Resources, D.G. and X.L.; Data Curation, D.G.; Writing—Original Draft Preparation, D.G.; Writing—Review & Editing, D.G. and X.L.; Visualization, D.G.; Supervision, X.L. and Z.X.; Project Administration, D.G.; Funding Acquisition, D.G., X.L. and Z.X. All authors have read and agreed to the published version of the manuscript.

**Funding:** This research received no external funding.

**Data Availability Statement:** The data presented in this study are available on request from the corresponding author. The data are not publicly available due to privacy.

**Acknowledgments:** The authors are grateful for the help from Chi Zhang and Xingyu Chen in collecting the elevator inventory and traffic data of the Tsinghua University campus. The authors also appreciate Beijing PARATERA Tech Co., Ltd. for providing the high-performance computational resources that were used in this work.

**Conflicts of Interest:** The authors declare no conflict of interest.

## Appendix A

**Table A1.** Ground motions selected from the NGA-West2 database.

| ID | Name | ID | Name |
|----|------|----|------|
| 1 | RSN9_BORREGO_B-ELC000.AT2 | 7 | RSN67_SFERN_ISD014.AT2 |
| 2 | RSN28_PARKF_C12050.AT2 | 8 | RSN76_SFERN_MA3130.AT2 |
| 3 | RSN40_BORREGO_A-SON033.AT2 | 9 | RSN84_SFERN_SDC000.AT2 |
| 4 | RSN51_SFERN_PVE065.AT2 | 10 | RSN86_SFERN_SON033.AT2 |
| 5 | RSN54_SFERN_BSF135.AT2 | 11 | RSN93_SFERN_WND143.AT2 |
| 6 | RSN55_SFERN_BVP090.AT2 | | |

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
