# Peer review of "Probability-Based City-Scale Risk Assessment of Passengers Trapped in Elevators under Earthquakes"

_sustainability, doi:10.3390/su15064829_

Round 1
Reviewer 1 Report
1) In point 3.1, the Authors distinguish models for different structures. It would increase understanding of the method if exemplary boundaries between different structures could be described. For example, when shear type deformation occurs and when flexural-shear coupling deformation takes place? In Figure 2 example of a tall building is presented, but above what height is the building considered high?
2) Parameter called fortification intensity (DI) is named (line 194). Please explain its meaning.
3) Figure 3 gives the probability of selecting an elevator type. Could the Authors support values of probability with any data? Why are all elevator candidates for older buildings given the same 25% probability and 50% probability for newer buildings? What is more, have all elevator candidates the same size (maximum number of passengers)?
4) In point 4.3, the Authors write that observation was carried out for 7 days. It would be interesting to see if the weather or the season influences people's behaviour, like the selection of elevators or stairs. Depending on different weather conditions, different numbers of PTEs can be observed, which can be crucial for the entire study.
Other issues:
1) All paper parts have incorrect numeration (lines 110, 156, 333, 507, 578).
2) All equations beside (2) are each twice numbered, some with wrong numbers.
3) It would be better to put lines 333 and 334 (Case study and Study area) on top of another one (10th page).
4) Caption of Figure 7 is separated from this Figure.
Reviewer 2 Report
The paper proposes and discusses the application of a probabilistic framework to assess the number of passengers trapped in elevators. The framework is based on the use of a simplified method previously proposed by the authors (city-scale Time-History Analysis) and the Monte Carlo procedure to predict the response of buildings in terms of engineering demand parameters. Then, by combining the predicted response of buildings, existing damage state functions for elevators, and elevator traffic statistics, the authors predict the number of passengers trapped in elevators by adopting as a case-study the campus of the Tsinghua University. Both a scenario-based assessment and an intensity-based assessment are performed showing the limitations and the advantages of the proposed procedure.
A revision of the English language is suggested.
1) Some suggestion about the English language and/or organization of the paper are indicated in the attached pdf file. Please consider reviewer’s notes.
2) Please avoid acronyms in the abstract (e.g., PTEs).
3) The term ridership is improper. Generally is used the term “passenger” referred to elevators, please check this along the entire paper.
4) When using the name of the authors to reference a work (e.g., Wang et al. page 2 row 51), please add the year of publication (Wang et al. 2017).
5) The number of each section is “1” (e.g. 1.Methodologies – 3.1 Probability-based city-scale THA). Possibly the section counter have some problems, please check.
6) Page 2 rows 63-65. The “second challenge” is clarified in the following text, but this sentence is too generic and not very clear. Please rewrite this sentence.
7) Page 2 rows 92-94. This sentence is not clear, please rewrite.
8) Section 2 – Workflow – Page 3 rows 111-117. This sentence seems to be hastily written, please consider to rewrite it.
9) Figure 1 and page 3 row 131. Key parameters for simulation should be clarified in this section or it should be indicated that will defined in §3.1 (Ωy and Ωp).
10) In Figure 1 and 3 the term “possibility” should changed with “probability”.
11) Some Equation references appear with repetition (e.g. (1) (1)). Further, after Equation 8, two numbers appear for each equation ((8)-(9)).Please check
12) Section 3- The authors differentiate shear and flexural-shear models based on the building height and the height-to-width ratio. For readers it could be interesting that threshold values are provided for this differentiation.
13) Page 5 – rows 173-197. The authors summarizes the four steps required to generate the simplified flexure-shear model. Since it is only a brief recall of an existing procedure, the procedure itself (or each step) shoud be referenced in the section.
14) Page 5 – row 194 (and Page 10 row 338). What the authors mean for “fortification intensity”?
15) Page 6 – rows 202-207: Despite it is clear the general meaning of the section, this should be rewritten considering the following flow: (i) at the bulding level such an information (number, type, installation year) is generally available, BUT at the large scale (ii) these information is generally lacking THUS (iii) a procedure is proposed to retrieve type of elevator and installation year.
16) Page 7 – row 241: the story acceleration is used as EDP to predict elevator damage. However, a brief recap when discussing EDPs in previous sections should be distinguish between the drift-sensitive and acceleration-sensitive components, that the proposed city-scale THA can predict both of them, but that this work is limited to the prediction of story acceleration since elevators are assumed to be acceleration-sensitive components.
17) Along the text the term “damage state” is used several times, but it is not clear if the authors refers to different damage states or consider only one damage state corresponding to the loss of functionality for the elevator. This should be clarified along the text. Further, a brief sentence should be added to evidence, according to authors opinginon, why FEMA fragilities can be adopted for the elevators in China.
18) Page 8 rows 293-300. This discussion about the approximation introduced during the estimation of the number of elevator riders, should be placed at the beginning of the section, then stating that a simplification will be adopted in this work.
19) Section 3 and 3.4 It is not clear if the number of simulation is set equal to 1000 for each building complex or is the total number of simulations. In section 3 it seems that is the total, but in 3.4 both options are considered. Further, in 3.4 it is recalled “the whole campus” anticipating Section 5, maybe it would be better to say “ at the large scale (see case-study in §5)”
20) Equation 5: it is not clear if RT is the “total number of people entering and exiting during the same period” from the elevator or the building. In the first case ETground,typical is equal to 1 otherwise the RT inclused people using the elevator or stairs. Please clarify.
21) Figure 11: Please indicate with (a),(b),(c),(d) for 8,12.30, 18.30, 22 – Similarly in Figure 13-14
22) Section 5: A target response spectrum is used to select spectrum-compatible ground motion records. Does it correspond to Ultimate Limit State, Damage Limite State? According to which code or study the spectrum has been chosen? Is it an Uniform Hazard Spectrum? Please add some details.
23) Both in $4.4 and $4.5 it is indicated which Gate would be prefereable for emergency forces. Which criterion has been used? Only because are the more closer?
24) Results in §4.4 and §4.5 show the numbers of Passengers Trapped in Elevator for scenario and intensity-based assessment. However, before to arrive to the final result, it would be also interesting to plot the probability for each elevator/building of reaching functionality limit state for different scenarios.
25) Page 18. Rows 562-577: Limitation for the proposed methodology could be numbered (e.g., i, ii,..)
